# Probabilistic Weight Fixing: Large-scale training of neural network weight uncertainties for quantization

**Christopher Subia-Waud**
School of Electronics & Computer Science
University of Southampton
Southampton, UK
cc2u18@soton.ac.uk

**Srinandan Dasmahapatra**
School of Electronics & Computer Science
University of Southampton
Southampton, UK
sd@ecs.soton.ac.uk

## Abstract

Weight-sharing quantization has emerged as a technique to reduce energy expenditure during inference in large neural networks by constraining their weights to a limited set of values. However, existing methods often assume weights are treated solely based on value, neglecting the unique role of weight position. This paper proposes a probabilistic framework based on Bayesian neural networks (BNNs) and a variational relaxation to identify which weights can be moved to which cluster center and to what degree based on their individual position-specific learned uncertainty distributions. We introduce a new initialization setting and a regularization term, enabling the training of BNNs with complex dataset-model combinations. Leveraging the flexibility of weight values from probability distributions, we enhance noise resilience and compressibility. Our iterative clustering procedure demonstrates superior compressibility and higher accuracy compared to state-of-the-art methods on both ResNet models and the more complex transformer-based architectures. In particular, our method outperforms the state-of-the-art quantization method top-1 accuracy by 1.6% on ImageNet using DeiT-Tiny, with its 5 million+ weights now represented by only 296 unique values. Code available at https://github.com/subiawaud/PWFN.

## 1 Introduction

Weight-sharing quantization is a technique developed to lower the energy costs associated with inference in deep neural networks. By constraining the network weights to take on a limited set of values, the technique can significantly reduce the data-movement costs within hardware accelerators, which represent the primary source of energy expenditure (DRAM read costs can be as much as 200 times higher than multiplication costs [Horowitz, 2014, Jouppi et al., 2017, Sze et al., 2017]). Storing the weight values close to computation and reusing them multiple times also becomes more feasible, thanks to the limited range of possible values. Various studies have explored the effectiveness of weight-sharing quantization, including [Ullrich et al., 2017, Subia-Waud and Dasmahapatra, 2022, Wu et al., 2018b, Han et al., 2015, 2016].

Weight-sharing quantization approaches face a common challenge in determining how much a single weight can be moved to a cluster center. Traditional methods rely on the magnitude or relative movement distances between weight and cluster to decide which weights can be moved to which clusters [Han et al., 2015, Subia-Waud and Dasmahapatra, 2022], irrespective of which filter or layer in the network the weight is located. However, this assumption neglects the fact that weights with the same value may have different roles in the network, and their placement within the architecture may affect their likelihood of being moved to a particular cluster. We posit that context-dependent movement of weights to clusters can, instead, better preserve the representational capacity of the

37th Conference on Neural Information Processing Systems (NeurIPS 2023).

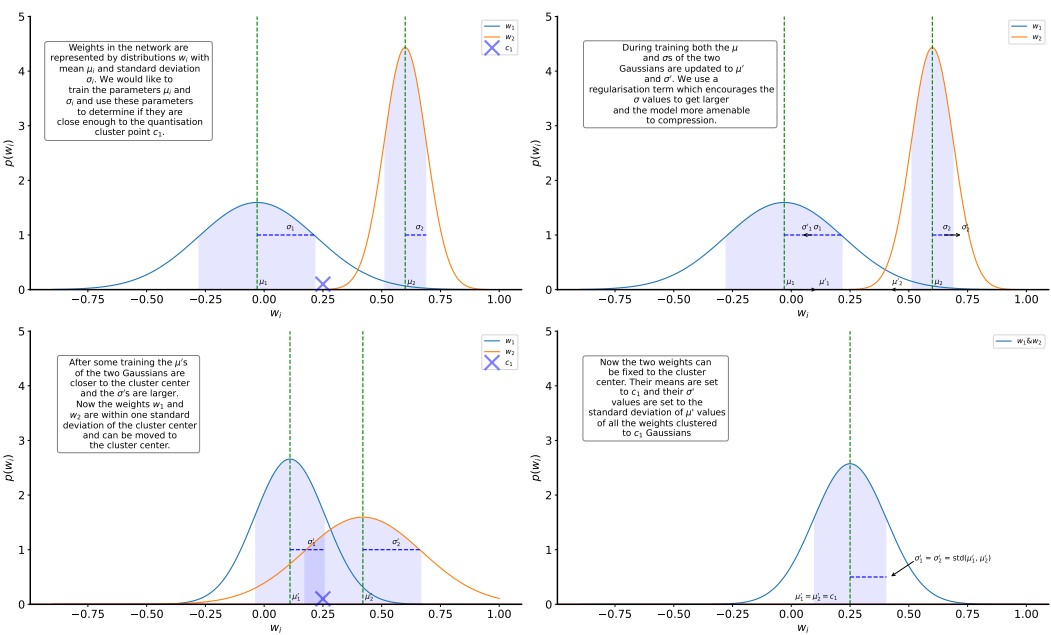

Figure 1: An overview of the PWFN process.

network while reducing its complexity. We build upon previous attempts [Wu et al., 2018b, Achterhold et al., 2018] to use a probabilistic framework to do so. The idea is to use a probability distribution to capture the flexibility in weight values, informing clustering decisions which reduce the entropy of the network and lower the unique parameter count without performance degradation.

We look to make progress on this problem with the use of Bayesian neural networks and a variational relaxation to identify optimal cluster configurations for the compression of neural networks with a method we call probabilistic weight fixing networks (PWFN). By incorporating the insights from the previous works on minimizing the relative weight-to-cluster distance [Subia-Waud and Dasmahapatra, 2022] and favoring additive powers-of-two cluster centroid values [Li et al., 2019b], we propose a novel initialization setting. Further, we discover that a simple regularization term which encourages maximal noise resilience is sufficient to avoid the variance of the weights' distribution from collapsing to zero. This regularization term facilitates the training of model-dataset pairings that were previously considered intractable when employing the variational Bayes-by-backdrop approach [Blundell et al., 2015]. These changes, when combined with a novel iterative clustering procedure, enable us to achieve superior compressibility. We can represent ResNet family models trained on the ImageNet dataset with fewer unique parameters and a reduced weight-space entropy while maintaining higher accuracy than the current state-of-the-art. Furthermore, we demonstrate the effectiveness of applying PWFN to transformer-based architectures, again achieving state-of-the-art results.

**Weight-sharing quantization.** Consider a neural network that consists of $N$ weights $\boldsymbol{w} = \{w_1, w_2, \ldots, w_N\}$. Each weight is typically unique and can take up to $2^b$ distinct values, where $b$ is the bit-width of the number system used. However, this poses a challenge since each time a weight is used, at least one memory read is required. Since memory reads dominate the computational cost of inference, it is desirable to reduce them. Weight-sharing quantization addresses this challenge using a reduced pool of cluster centers $\boldsymbol{c} = \{c_1, c_2, \ldots, c_k\}$, with $k \ll N$ and defining a map $\boldsymbol{w} \to \boldsymbol{c}$, where each weight $\boldsymbol{w}_i \in \boldsymbol{w}$ is mapped to a cluster center: $w_i \mapsto c_k \in \boldsymbol{c}$.

Two considerations are pivotal for formulating this process: determining which values should be included in the reduced cluster pool $\boldsymbol{c}$, and identifying the mapping of weights in $\boldsymbol{w}$ to the clusters in $\boldsymbol{c}$.

**Which values should be in the cluster pool?** Insights from the literature highlight that pre-trained weight distributions often adhere to a heavy-tailed distribution [Barsbey et al., 2021, Hodgkinson and Mahoney, 2021, Gurbuzbalaban et al., 2021]. Moreover, hardware implementations show a preference for powers-of-two values for weights due to their cost-effective multiplication properties

[Vogel et al., 2019, Przewlocka-Rus and Kryjak, 2023, Lee et al., 2017]. While some methods advocate for exclusive use of powers-of-two as cluster centers [Zhou et al., 2017], others propose a more flexible approach, suggesting additive powers-of-two [Li et al., 2019b, Subia-Waud and Dasmahapatra, 2022] – but only if it doesn't compromise performance. In our study, we align with this perspective, populating the cluster pool with powers-of-two values, but making exceptions for additive powers-of-two when they enhance performance.

**How to identify which of the weights should be moved to which cluster?** Previous studies have shown that distance metrics can be utilized to determine which fixed clusters can accommodate certain weights without negatively affecting the performance. Weight-sharing clustering techniques can rely on these metrics to assign weights to clusters. The commonly used distance metrics include Euclidean distance [Wu et al., 2018b] and Relative movement distance [Subia-Waud and Dasmahapatra, 2022]. However, these metrics implicitly assume that all weights with the same value should be treated equally, irrespective of their location in the network. This may not be valid as moving a small (large) weight by a small (large) distance affects classification outcome depending on where the weight is in the network. To overcome this limiting assumption, we apply in this paper a Bayesian neural network (BNN) training approach to obtain a metric to determine the allowed movement of weights to clusters.

## 2   Related Work

**Bayesian Neural Networks.** BNNs attempt to use the methods of Bayesian inference in modeling predictive problems. Rather than the weights in a network coming from point estimates (i.e., a single value for each weight), a BNN attempts to model many (ideally all) configurations of weight values throughout a network and make predictions, weighting each configuration by its probability. Exact Bayesian inference on the weights would require the computation of the integral $P(\boldsymbol{y}|x, D) = \int P(\boldsymbol{y}|x, \boldsymbol{w})P(\boldsymbol{w}|D)d\boldsymbol{w}$ where predictions for each allowed $\boldsymbol{w}$ are averaged over. Unfortunately, the marginalization over $P(\boldsymbol{w}|D)$ is intractable for even simple networks, so approximations are needed. Approaches to this include Laplace approximation [MacKay, 1992], gradient MCMC [Welling and Teh, 2011], expectation propagation updates [Hernández-Lobato and Adams, 2015], and variational methods such as Bayes-by-backprop (BBP) [Blundell et al., 2015]. Using dropout at inference time has also been shown to be a variational approximation [Gal and Ghahramani, 2016] which is much more amenable to scale – albeit with reduced expressive power [Jospin et al., 2022]. BBP, of which our approach uses a variation of, models each of the network weights as coming from a Gaussian distribution and uses the re-parameterization trick for gradient descent to learn the parameters of the weight distributions. We give the derivations to do so in the Appendix. BBP has been demonstrated to work well in more complex model-dataset combinations than other BNN approaches (aside from dropout) but is still not able to scale to modern architectures and problems such as the ImageNet dataset and Resnet family of networks [Jospin et al., 2022].

One reason for the limited applicability is that the weight prior, which is a scaled mixture of two zero-mean Gaussians [Blundell et al., 2015], is not a well-motivated uninformative prior [Fortuin, 2022]. Instead, we initialize our training from a pre-trained network, in the spirit of empirical Bayes, and propose a prior on weight variances that encourages noise-resilience. A noise-resilient network can be thought of as one with a posterior distribution with large variances at its modes for a given choice of weight coordinates. Thus, small changes in weight space do not meaningfully alter the loss. The corresponding characterization of the flat minima in the loss landscape is well supported in the optimization literature [Foret et al., 2020, Kaddour et al., 2022].

**Quantization.** Quantization is a technique used to reduce the number of bits used to represent components of a neural network in order to decrease energy costs associated with multiplication of 32-bit floating-point numbers. There are different approaches to quantization, such as quantizing weights, gradients, and activations. Clipping+scaling quantization maps the weight $w_i$ to $w_i' = \text{round}(\frac{w_i}{s})$, where $\text{round}()$ is a predetermined rounding function and $s$ is a scaling factor learned channel-wise, layerwise or with even further granularity. Quantization can be performed post-training or with quantization-aware training [Jacob et al., 2018].

Weight sharing quantization uses clustering techniques to group weights and fix the weight values to their assigned cluster centroid. These weights are stored as codebook indices, which enables compressed representation methods like Huffman encoding to compress the network further. Unlike

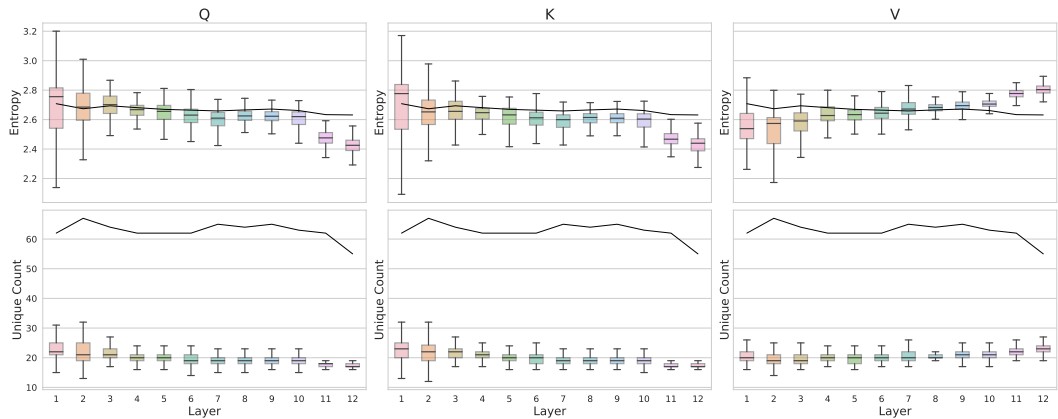

Figure 2: For DeiT small, we show a box plot of the entropies and unique counts per input channel for each Q,K,V by layer and with the mean of each layer (calculated across all attention heads) shown in the black lines.

clipping+scaling quantization, weight sharing techniques share the pool of weights across the entire network. Several studies have proposed different methods for weight sharing quantization. For example, Zhou et al. [Zhou et al., 2017] restrict layerwise rounding of weights to powers-of-two for energy-cheap bit-shift multiplication, while [Li et al., 2019a] suggest additive powers-of-two (APoT) to capture the pre-quantized distribution of weights better. [Wu et al., 2018a] use a spectrally relaxed k-means regularization term to encourage the network weights to be more amenable to clustering and focus on a filter-row codebook for convolution. Similarly, [Stock et al., 2020] and [Fan et al., 2020] focus on quantizing groups of weights into single codewords rather than the individual weights themselves. [Ullrich et al., 2017] use the distance from an evolving Gaussian mixture as a regularization term to prepare the clustering weights. However, their approach is computationally expensive. In contrast, our formulation reduces computation by adding cluster centers iteratively when needed and only considering weights that are not already fixed for a regularization prior. A work closely related to ours is that of [Achterhold et al., 2018], where the authors formulate post-training quantization and pruning as a variational inference problem with a 'quantizing prior' but due to its optimization complexity and difficulties with weights not being drawn tightly enough into clusters the method was only demonstrated to work on simple dataset-model combinations and for the case of ternary quantization. WFN [Subia-Waud and Dasmahapatra, 2022] is a recent weight-sharing approach that uses the minimizing of the relative movement distance to determine which weights are to be clustered and has demonstrated that a low weight-space entropy and few unique weights with a single codebook can maintain accuracy. In our work, we go much further in reducing weight-space entropy and unique weight count by using a context-aware distance metric and probabilistic framework in determining which weights can be moved where. Additionally, almost all of the works reviewed do not quantize the first and last layers [Li et al., 2019a, Jung et al., 2019, Zhou et al., 2016, Yamamoto, 2021, Oh et al., 2021, Li et al., 2022] in order to maintain performance and in some cases don't quantize the bias terms [Li et al., 2022], - we challenge this view and attempt a full network quantization, leaving only the batch-norm and layer-norm parameters at full precision.

## 3 PWFN

In PWFN, we follow $T$ fixing iterations each of which combines a training and a clustering stage in order to reach an outcome of a highly compressed/quantized network with a single whole-network codebook. We model each individual weight $w_i$ as a draw from a distribution with learnable parameters and use these parameters as guidance for the clustering stage. We model each $w_i \in \boldsymbol{w}$ as coming from a Gaussian distribution $\mathcal{N}(\mu_i, \sigma_i)$ and during the training stage we use a form of BBP to train the weight distribution parameters $\boldsymbol{\mu} = (\mu_1, \ldots, \mu_N)$ and $\boldsymbol{\sigma} = (\sigma_1, \ldots, \sigma_N)$ to minimize both the task performance loss and to encourage the weight distributions to tell us exactly how much noise can be added to each weight without affecting performance. Both $\boldsymbol{\mu}$ and $\boldsymbol{\sigma}$ are trained with an additional regularization term that encourages larger values of $\boldsymbol{\sigma}$ to counter the model reverting to the point estimates with $\sigma_i = 0$ to minimize the classification loss. During the clustering stage, we look

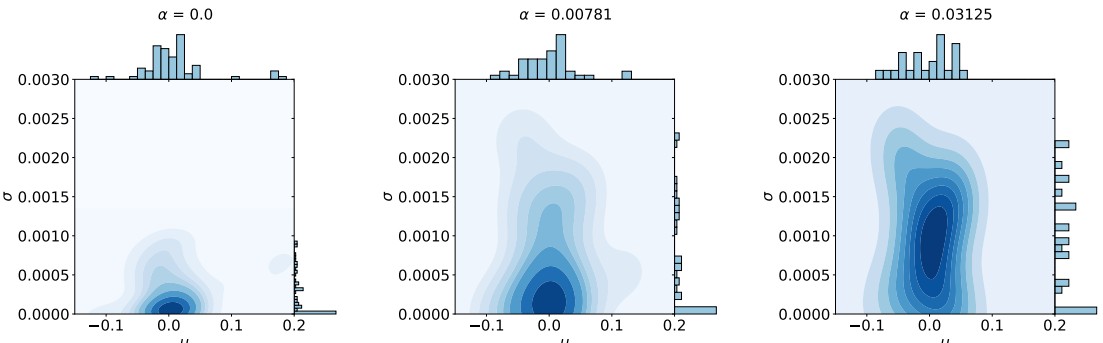

Figure 3: The regularization term acts to stop the $\sigma$ uncertainty values from collapsing to zero. This experiment is run using the Cifar10 dataset with ResNet-18, stopping after 30 epochs.

to use this information to move the $\mu_i$ values to one of a handful of cluster centers. We favour the cluster centers to be hardware multiplication-friendly powers-of-two, or additive-powers-of-two. After $T$ iterations of training and clustering, each of the weights' distributions in the networks will have their $\boldsymbol{\mu}$ values centered on one of the $k$ clusters $\boldsymbol{c}$ in the codebook.

After the $T$ fixing iterations there are two options depending on the downstream usage of the network: either the network can be converted into point estimates and the weights set to the exact $\mu$ values giving us a quantized network. Or, we can use the extra information given to us by modelling each weight as a distribution as a way of quantifying uncertainty of a particular prediction. If, after multiple samples of $\boldsymbol{w}$, a model changes its prediction for a fixed input, this tells us that there is uncertainty in these predictions – with this information being useful for practical settings.

**PWFN Training.** Consider a network parameterized by $N$ weights $\boldsymbol{w} = \{w_1, ..., w_N\}$. In PWFN, each weight $w_i$ is not a single value but is instead drawn from a distribution $w_i \sim \mathcal{N}(\mu_i, \sigma_i)$, and instead of learning the $w_i$ directly, the learning process optimizes each $\mu_i$ and $\sigma_i$. In a forward pass, during training, we sample weight values $w_i$ according to its distribution:

$$w_i = \mu_i + \sigma_i \epsilon, \ \epsilon \sim \mathcal{N}(0, 1). \tag{1}$$

The forward pass is stochastic under fixed $\boldsymbol{\mu}, \boldsymbol{\sigma}$. If trained correctly, the $\sigma_i$ values give us information about the amount of noise a particular weight $w_i$ can handle without affecting performance. Said another way, if we can find a configuration $\boldsymbol{w} = (\boldsymbol{\mu}, \boldsymbol{\sigma})$ which maintains task performance despite the randomness introduced by the $\sigma_i$ parameters, then we will know which of the corresponding weights can be moved and to what degree. In PWFN, we train $\boldsymbol{\mu}, \boldsymbol{\sigma}$ following the BBP optimization process [Blundell et al., 2015] with some changes both in terms of initialization and the priors on $\boldsymbol{\mu}$ and $\boldsymbol{\sigma}$.

**Large $\sigma$ constraint for $\boldsymbol{w}$.** Given the usual cross-entropy or other performance loss, there is a clear direction of travel during gradient descent towards having small $\sigma_i$ values and less uncertainty in the network parameters. A prior on the distribution of weights is therefore needed to prevent the $\boldsymbol{\sigma} = 0$ point estimate solution being found which would leave us with no weight movement information. In the original BBP set-up, the authors looked to prevent vanishing variance by regularising the distribution of weights according to a prior distribution of a mixture of zero-mean Gaussian densities with different variances (the parameters of the prior they find through an exhaustive search). The motivation for doing so was because the empirical Bayes approach didn't perform well due to the network favoring updating these parameters over the posterior (since there are fewer) and the link to the successful spike-and-slab prior [Mitchell and Beauchamp, 1988] — where values are concentrated around 0 (the *slab*) or another value known as the *spike* – favoring sparsity. Instead, we hypothesize that a *good* network can handle the most noise injection whilst maintaining performance. These networks are likely more compressible since they have been trained to accept changes to their weight values without performance degradation during training.

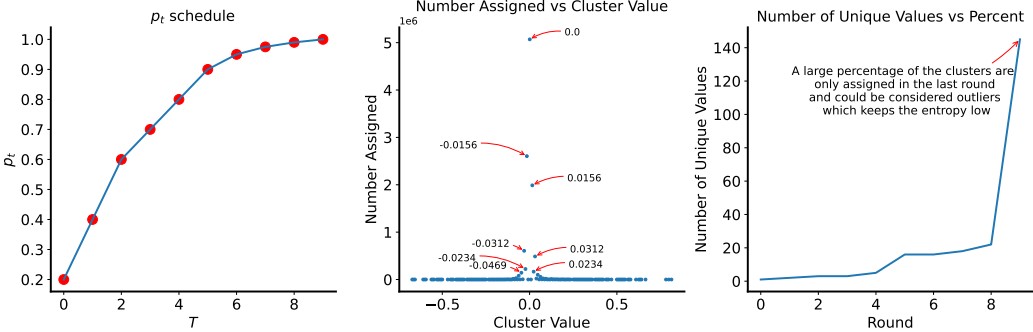

Figure 4: $p_t$ follows the same schedule as [Subia-Waud and Dasmahapatra, 2022] (left). In the middle and right plots, we see that PWFN achieves very small entropy values by majority of weights to only a very small (4 or 5) cluster values and the rest are assigned as outliers, most of which are powers-of-two.

We attempt this by making our $\boldsymbol{\sigma}$ values to be large. Networks with large $\boldsymbol{\sigma}$ have, probabilistically, more noise added to the $\boldsymbol{\mu}$ values during training and so have to learn to have robust performance under such circumstances. We note that this acts as a push-pull relationship with the performance loss, which favours low $\boldsymbol{\sigma}$ values. The motivation is that, much like $L_1$ norms enforcing sparsity, this formulation will train the network to produce a large $\sigma_i$ for noise-resilient parameter $w_i$, whilst maintaining a noise-sensitive weight $w_j$ to have a small $\sigma_j$ despite the prior pull. The regularised loss function for training the training phases of the algorithm is:

$$- \log P(\mathcal{D}|\boldsymbol{\mu}, \boldsymbol{\sigma}) + \alpha \mathcal{L}_{\text{REG}}(\boldsymbol{\sigma}), \tag{2}$$

where the regularization term is:

$$\mathcal{L}_{\text{REG}}(\boldsymbol{\sigma}) = \sum_{i=1}^{N} \mathcal{L}(\sigma_i) = -\sum_{i=1}^{N}(\sigma_i - S)\Theta(S - \sigma_i), \tag{3}$$

with $\Theta(x) = 1$ for $x \geq 0$ and 0 otherwise. The $\Theta$ function prevents the optimization from finding a network with a subset of $\boldsymbol{\sigma}$ with infinitely large values and dominating the cross entropy term. $S$ is thus a cutoff on how large the values in $\boldsymbol{\sigma}$ can be. $\alpha$ is a hyperparameter controlling the formation of a noise-resilient network where the *majority* of the weights can receive noise injection without hurting performance, not just a few. In Figure 3 we illustrate the effect on the distribution of $\boldsymbol{\sigma}$ under different $\alpha$ values for a ResNet-18 trained on the Cifar-10 dataset. As we increase $\alpha$ the $\boldsymbol{\sigma}$ values no longer collapse to zero giving us information for downstream clustering.

**Initialization using Relative Distance from Powers-of-two.** For each weight $w_i$ we need to specify its prior distribution so as to derive the posterior using Bayesian updating. We assume that the posterior distribution is Gaussian with a diagonal covariance matrix: $P(w_i; \mu_i, \sigma_i)$ whose parameters $\mu_i, \sigma_i$ are trained using BBP. To initialize the prior distributions for $\mu_i$ and $\sigma_i$ we set $P^0(\mu) = \prod_i P^0(\mu_i)$ where $P^0(\mu_i) \propto \delta_{\mu_i, w_i}$ for the pre-trained weight value $w_i$. For a Gaussian posterior we would typically require an unknown $\sigma$ to be drawn from a Gamma conjugate prior distribution. Instead, we set $\sigma_i$ to be a known function of the $\mu_i$ at initialization. In [Subia-Waud and Dasmahapatra, 2022] relative distances to the preferred powers of two values for the weight was used to determine weight movement. To favour anchoring weights at powers of two, we set the standard deviations to be smallest $(2^{-30})$ at either edge of each interval between the nearest integer powers of two, $(2^{x_i} \leq \mu_i \leq 2^{x_i+1})$ for integer $x_i$, and largest at the midpoint of the interval. We introduce a parabolic function $\sigma_i(\mu_i)$ as a product of relative distances of each pre-trained weight value ($\mu_i$) to the nearest lower and upper powers of two:

$$\sigma_i(\mu_i) = (0.05)^2 \left(\frac{|2^{x_i} - \mu_i|}{|2^{x_i}|}\right)\left(\frac{|\mu_i - 2^{x_i+1}|}{|2^{x_i+1}|}\right), \tag{4}$$

(Full details and visualization is given in the appendix).

**PWFN Clustering.** In Figure 1 we show a schematic of the clustering stage in which we use the information garnered from the weights' distribution parameters to identify cluster centers and their

assignment. PWFN clustering is a two-step method running for $t = 1, \ldots, T$ iterations. At each step we set a fraction $p_t$ of the weights to be fixed, so that $|W_{\text{fixed}}^t| = Np_t$. The remaining weights at iteration stage $t$ are trainable and called $W_{\text{free}}^t$. We follow the scheme first proposed in [Subia-Waud and Dasmahapatra, 2022] in setting $p_t$ (Figure 4, left). All of the weights $w_i$ that are assigned to $W_{\text{fixed}}^t$ will have their $\mu_i$ values fixed to one of the set of cluster centers. At the last iteration, $|W_{\text{free}}^T| = 0$ and $p_T = 1$, as all weights have been fixed to their allocated cluster centroids.

We next introduce how a cluster center $c_k$ is defined and how the mapping $\mu_i \mapsto c_k \in \boldsymbol{c}$ is performed. Let $R = \{-\frac{1}{2^b}, \ldots, -\frac{1}{2^{j+1}}, -\frac{1}{2^j}, 0, \frac{1}{2^j}, \frac{1}{2^{j+1}}, \ldots \frac{1}{2^b}\}$ be the set of all powers-of-two up to a precision $b$. For a weight to be a desired additive power of two, a sum over at most $\omega$ elements of $R$ is defined to be a cluster center of order $\omega$. Formally, for $\mathcal{P}(R)$ the power set of $R$,

$$\boldsymbol{c}^\omega = \{\sum_{i \in r} i \mid r \in \mathcal{P}(R) \wedge |r| \leq \omega\}. \tag{5}$$

PWFN begins with order $\omega = 1$, the powers-of-two up to precision $b$ as the proposal cluster set $\boldsymbol{c}^\omega$. Next, for each weight $w_i = (\mu_i, \sigma_i)$ in the network, the value of $\sigma_i$ is used to determine how far away they are from each of the cluster centers using:

$$D_{\text{prob}}(w_i, c_j) = \frac{|\mu_i - c_j|}{\sigma_i}. \tag{6}$$

Interpret this Mahalanobis distance as: "how many sigmas (standard deviations) away is cluster $c_j \in \boldsymbol{c}^\omega$ from weight $w_i$". At iteration stage $t$, for each free weight we define $c_*^\omega(i) = \min_{c \in \boldsymbol{c}^\omega} D_{\text{prob}}(w_i, c)$ as the cluster center that is the fewest sigmas away from $w_i \in W_{\text{free}}^t$. We denote by $n_k^\omega$ the number of weights with the smallest $D_{\text{prob}}$ to cluster $c_k^\omega$, i.e., $n_k^\omega = \sum_i \mathbb{I}[c_k^\omega = c_*^\omega(i)]$. We then take the index $k^*$ of the cluster with the most number of weights nearest to a cluster: $k^* = \text{argmax}_k \, n_k^\omega$. Thus, $c_{k^*}^\omega \in \boldsymbol{c}^\omega = (c_1^\omega, \ldots, c_k^\omega)$ is the cluster with the most number of weights nearest to it.

We then order the weights in $W_{\text{free}}^t$ by their distance to $c_{k^*}^\omega$. In detail, for $W_{\text{free}}^t = [w_1, \ldots, w_i, \ldots, w_n]$, we reorder the weights by permuting the indices $w_i' = w_{\pi(i)}$ where $\pi : [1, \ldots, n] \to [1, \ldots, n]$ is a permutation, $i \mapsto \pi(i)$. The ordered list $[w_1', \ldots, w_n']$ satisfies

$$D_{\text{prob}}(w_i', c_{k^*}^\omega) \leq D_{\text{prob}}(w_{i+1}', c_{k^*}^\omega) \tag{7}$$

Next, we need to determine how many of these weights we should assign to cluster $c_{k^*}^\omega$. To do so, we define a threshold $\delta$ and we take the first $\ell(\delta)$ weights from $[w_1', \ldots, w_n']$ such that:

$$\frac{1}{\ell(\delta)} \sum_{i=1}^{\ell(\delta)} D_{\text{prob}}(w_i', c_{k^*}^\omega) \leq \delta. \tag{8}$$

As long as this is possible with $\ell(\delta) > 0$, we have identified both a cluster $c_{k^*}^\omega$ and set of weights $[w_1', \ldots, w_{\ell(\delta)}']$ which can be moved from $W_{\text{free}}^t$ to $W_{\text{fixed}}^{t+1}$. We map the weights in $[w_1', \ldots, w_{\ell(\delta)}'] = [(\mu_1', \sigma_1'), \ldots, (\mu_{\ell(\delta)}', \sigma_{\ell(\delta)}')]$ to a single weight $w_{k^*} = (\mu_{k^*}, \sigma_{k^*})$ corresponding to cluster $c_{k^*}^\omega$: $\mu_{k^*} = c_{k^*}^\omega$ and $\sigma_{k^*} = \texttt{std}([\mu_1', \ldots, \mu_{\ell(\delta)}'])$ where $\texttt{std}$ computes the standard deviation of its argument. This process is then repeated, finding the next most popular cluster until $Np_t$ weights are assigned a cluster. If $\ell(\delta) = 0$, before enough weights are assigned in iteration $t$, then we have not been able to find any cluster centers $c_j \in \boldsymbol{c}^\omega$ which are close enough to any weight, i.e., $D_{\text{prob}}(w_i, c_j) > \delta$ for all weights $w_i \in W_{\text{free}}^t$ and $c_j = c_{k^*}$. In this case, we set $\omega \leftarrow \omega + 1$ and $\delta \leftarrow 2\delta$ giving us a higher-order additive powers-of-two set and less restrictive $\delta$ value threshold. Since $|\boldsymbol{c}^{\omega+1}| > |\boldsymbol{c}^\omega|$, this increase in $\omega$ makes more cluster centers available during the next clustering attempt.

**Putting it All Together.** Putting the training and clustering stages together, we have a process for training a neural network whose weights are from a Gaussian posterior distribution with diagonal covariance matrix by backpropagation (BPP) that favours configurations with long Gaussian tails, which the clustering stage can then use to identify which weights to move and to what extent. This process is repeated for $T$ iterations, with the fraction $p_t$ of weights increasing with each $t$ $p_{t+1} > p_t$ until all of the weights are moved from $W_{\text{free}}$ to $W_{\text{fixed}}$ at iteration $T$ where $p_T = 1$. We give the full algorithm in the Appendix.

Table 1: Full comparison results. (w/o FL-Bias) refers to calculating the metrics without the first-last layers and bias terms included. 'Params' refers to the unique parameter count in the quantized model, entropy is the full weight-space entropy. In-ch, layer, attn refer to whether the method uses a separate codebook for each layer, in-channel and attention head respectively.

| Model | Method | Separate Codebook | | | Top-1 (Ensemble) | Entropy | Params |
| | | Layer | In-ch | Attn | | | |
|---|---|---|---|---|---|---|---|
| ResNet-18 | Baseline | - | - | - | 68.9 | 23.3 | 10756029 |
| | LSQ | ✓ | ✗ | - | 68.2 | - | - |
| | APoT | ✓ | ✗ | - | 69.9 | 5.7 | 1430 |
| | WFN | ✗ | ✗ | - | 69.7 | 3.0 | 164 |
| | PWFN (no prior) | ✗ | ✗ | - | 69.3 (69.6) | **1.7** | **143** |
| | PWFN | ✗ | ✗ | - | **70.0 (70.1)** | 2.5 | 155 |
| ResNet-34 | Baseline | - | - | - | 73.3 | 24.1 | 19014310 |
| | LSQ | ✓ | ✗ | - | 71.9 | - | - |
| | APoT | ✓ | ✗ | - | 73.4 | 6.8 | 16748 |
| | WFN | ✗ | ✗ | - | 73.0 | 3.8 | 233 |
| | PWFN (no prior) | ✗ | ✗ | - | 73.5 (74.4) | **1.2** | **147** |
| | PWFN | ✗ | ✗ | - | **74.3 (74.6)** | 1.8 | 154 |
| ResNet-50 | Baseline | - | - | - | 76.1 | 24.2 | 19915744 |
| | LSQ | ✓ | ✗ | - | 75.8 | - | - |
| | WFN | ✗ | ✗ | - | 76.0 | 4.1 | **261** |
| | PWFN (no prior) | ✗ | ✗ | - | 77.2 (78.1) | 3.5 | 334 |
| | PWFN | ✗ | ✗ | - | **77.5 (78.3)** | **3.4** | 325 |
| DeiT-Small | Baseline | - | - | - | 79.9 | 16.7 | 19174713 |
| | LSQ+ | ✓ | ✓ | ✗ | 77.8 | - | - |
| | Q-ViT | ✓ | ✓ | ✓ | **78.1** | 11.3 | 3066917 |
| | Q-ViT (w/o FL-Bias) | ✓ | ✓ | ✓ | **78.1** | 10.4 | 257149 |
| | PWFN (no prior) | ✗ | ✗ | ✗ | 78.0 (78.3) | **2.7** | **352** |
| | PWFN | ✗ | ✗ | ✗ | **78.1 (78.5)** | **2.7** | 356 |
| DeiT-Tiny | Baseline | - | - | - | 72.9 | 15.5 | 5481081 |
| | LSQ+ | ✓ | ✓ | ✗ | 68.1 | - | - |
| | Q-ViT | ✓ | ✓ | ✓ | 69.6 | 11.5 | 1117630 |
| | Q-ViT (w/o FL-Bias) | ✓ | ✓ | ✓ | 69.6 | 10.5 | 128793 |
| | PWFN (no prior) | ✗ | ✗ | ✗ | **71.4 (71.6)** | **2.8** | 300 |
| | PWFN | ✗ | ✗ | ✗ | 71.2 (71.5) | **2.8** | **296** |
| DenseNet161 | Baseline | - | - | - | 77.8 | 17.1 | 26423159 |
| | PWFN | ✗ | ✗ | ✗ | 77.6 (78.0) | 1.1 | 125 |

# 4   Experiments

We conduct our experimentation on the ImageNet dataset with a wide range of models: ResNets-(18,34,50) [He et al., 2016], DenseNet-161 [Huang et al., 2017] and the challenging DeiT (small and tiny) [Touvron et al., 2021]. For each model, we convert all the parameters in the convolution and linear layers into Gaussian distributions where the mean value is set to be the weight value of the pre-trained model found in the Timm library. Thus, at test time with no further training, we retain the original accuracies. We set the variance parameters according to the setting described in Eq (12). We then apply nine rounds of the described weight fixing with three epochs of re-training each round, totalling to 27 epochs of training. We train using SGD with momentum 0.9 with a learning rate of 0.001. For all experiments, we fix $\delta = 1$, $\alpha = 2^{-11}$ which we found using grid-search on the Cifar-10 dataset and works surprisingly well in all settings. For all our experiments we train using 4x RTX8000 GPUs and a batch-size of 128. For the ensemble results, we sample 20 times different weights using the learned weights' distributions and report the mean accuracy.

# 5   Results

We compare PWFN against a range of quantization approaches where the model weights have been made available so that we can make accurate measurements of entropy and unique parameter count.

Table 2: Comparison of the number of additional training epochs required by different fine-tuning quantization methods.

| Method | Num of additional epochs |
|--------|:---:|
| ApoT | 120 |
| PWFN | 27 |
| WFN | 27 |
| LSQ | 90 |
| QviT | 300 |

For the ResNet family, we compare against the current state-of-the-art APoT [Li et al., 2019b] [1] and WFN [Subia-Waud and Dasmahapatra, 2022] [2]. For the transformer models, there has only been one work released, Q-Vit [Li et al., 2022] [3], which has both the model saves and code released. For both APoT and Q-Vit, we compare the 3-bit models which are the closest in terms of weight-space entropy to that achieved by PWFN.

As presented in Table 2, PWFN requires substantially fewer additional training epochs than most methods, save for WFN, highlighting its training efficiency. We use a straightforward regularization term that encourages an increase in $\sigma$, and its computational cost is comparable to that of l1 regularization. While our approach does lead to greater memory demands due to the additional $\sigma$ parameters and their associated gradient updates, the overall simplicity of the method is more efficient than previous BNN training procedures making it feasible to tackle more complex model-dataset pairings. Additionally, we note that when using the quantized version for inference, there are no extra costs, and the BNN functions as a point-estimate network.

In Table 1 we can see the set of results. PWFN demonstrates superior entropy, unique parameter count and top-1 accuracy across the board. In addition to the point-estimate accuracy using the mean of each of the weights' distributions (the cluster centers), we can additionally sample the weights from the learned distributions to give us an ensemble of models the mean prediction of which gives us further accuracy gains which we show in brackets in the Table. The prior initialization gives a slight but consistent accuracy improvement over using a uniform prior (PWFN (no prior)). We note that for both APoT and Q-Vit, different codebooks are used for different layers and for Q-Vit, different codebooks were additionally used for every attention head and input channel, and the bias terms were left unquantized, pushing up the parameter count and weight-space entropy substantially. We highlight this as a growing trend in the field, where relaxations such as leaving large parts of the network unquantized, or using different codebooks for ever granular parts of the network, are often used. Each relaxation comes at a cost in hardware, be that support for unquantized elements – such as the first and last layers – or the use of different codebooks for various parts of the architecture. Figure 2 illustrates the variation in entropy and the count of unique parameters across different layers and attention components. A notable observation from our study is that the weights associated with the 'value' component exhibit higher entropy in the final layer. This observation aligns with the notion that employing a fixed quantization scheme for each layer necessitates a relaxation of the quantization constraints specifically for the last layer, as supported by prior studies [Li et al., 2019a, Jung et al., 2019, Zhou et al., 2016, Yamamoto, 2021, Oh et al., 2021, Li et al., 2022]. Moreover, this highlights an intriguing possibility that in the context of attention networks, such relaxation might be essential only for the 'value' weights, and not for the 'keys' and 'queries'.

In understanding how PWFN is able to compress a network's representation to such a degree compared to WFN we look to how often the previously proposed relative distance threshold is maintained.

In Figure 5, it's evident that while the relative distance threshold established in WFN is, on average, maintained, there are edge-cases where it isn't. This observation suggests that having a context-specific noise tolerance benefits subsequent compression stages. Furthermore, the data indicates that these values are typically small (as seen in the left column), have a high frequency of occurrence (depicted in the middle), and are predominantly assigned during the middle (0.6, 0.7) and final rounds.

---

[1] https://github.com/yhhhli/APoT_Quantization
[2] https://github.com/subiawaud/Weight_Fix_Networks
[3] https://github.com/YanjingLi0202/Q-ViT

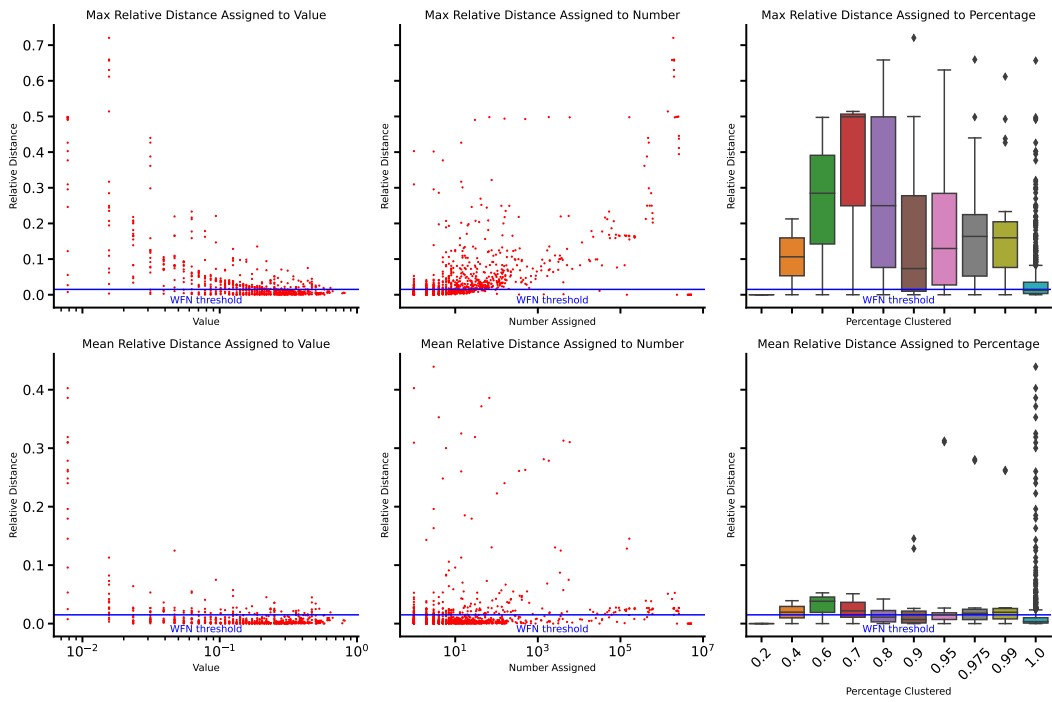

Figure 5: The maximum (top left) and mean (bottom left) relative distance a weight moves to a cluster by cluster value. The maximum relative distance is not well maintained with the number of weights assigned to that cluster (top middle), but the mean relative distance is (bottom middle). The maximum (top) relative distance for each cluster assignment and mean (bottom) relative distance by round are shown in the right-hand column. In all plots, we show in blue the threshold used in WFN.

# 6 Conclusion

This work proposed PWFN, a training and clustering methodology that can both scale BNNs to complex model-dataset combinations and then use the weight uncertainties to inform quantization decision-making. The result is a set of networks with extremely low weight-space entropies that can be used downstream in accelerator designs to mitigate expensive data movement costs. Additionally, we have seen the potential of the probabilistic aspect of the learned networks with a sampled ensemble giving noticeable accuracy gains. An exciting direction for future work is to explore how the uncertainty estimations and the out-of-distribution performance of neural networks could be enhanced using PWFN to train Bayesian Neural Networks.

# 7 Acknowledgments

This work was supported by the UK Research and Innovation Centre for Doctoral Training in Machine Intelligence for Nano-electronic Devices and Systems [EP/S024298/1]

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

# 8 Appendix

**Bayes-by-backprop and PWFN** The Bayesian posterior neural network distribution $P(\boldsymbol{w}|\boldsymbol{D})$ is approximated by a distribution $Q(\boldsymbol{w}|\theta)$ whose parameters $\theta$ are trained using back-propagation, Bayes-by-backprop (BPP). The approximation is achieved by minimizing the Kullback-Leibler (KL) divergence $D_{KL}[Q\|P]$ between $P$ and $Q$ to find the optimal parameters $\theta^*$. These parameters $\theta^*$ instantiate the means $\mu_i$ and variances $\sigma_i^2$ of the PWFN.

$$\theta^* = \arg\min_\theta KL[Q(\boldsymbol{w}|\theta)\|P(\boldsymbol{w}|\boldsymbol{D})], \text{ where}$$

$$KL[Q(\boldsymbol{w}|\theta)\|P(\boldsymbol{w}|\boldsymbol{D})] = \mathbb{E}_{Q(\boldsymbol{w}|\theta)} \log\left(\frac{Q(\boldsymbol{w}|\theta)}{P(\boldsymbol{w}|\boldsymbol{D})}\right) = \mathbb{E}_{Q(\boldsymbol{w}|\theta)} \log\left(\frac{Q(\boldsymbol{w}|\theta)P(\boldsymbol{D})}{P(\boldsymbol{D}|\boldsymbol{w})P(\boldsymbol{w})}\right). \tag{9}$$

The $P(\boldsymbol{D})$ term does not contribute to the optimization and is dropped, leaving

$$\begin{aligned}\theta^* &= \arg\min_\theta \mathbb{E}_{Q(\boldsymbol{w}|\theta)}[\log Q(\boldsymbol{w}|\theta) - \log P(\boldsymbol{D}|\boldsymbol{w}) - \log P(\boldsymbol{w})], \\ &\approx \arg\min_\theta \sum_m \underbrace{\log Q(\boldsymbol{w}^m|\theta) - \log P(\boldsymbol{w}^m)}_{\text{prior dependent}} - \underbrace{\log P(\boldsymbol{D}|\boldsymbol{w}^m)}_{\text{data dependent}},\end{aligned} \tag{10}$$

where the expectation value is approximated by samples $\boldsymbol{w}^m \sim Q(\boldsymbol{w}|\theta)$ drawn from $Q(\boldsymbol{w}|\theta)$ each of which instantiates a neural network.

Gradient descent over each $\boldsymbol{w}^m$ that instantiates a neural network is made possible by the *reparametrization trick* [Kingma and Welling, 2013, Blundell et al., 2015]. The idea is to regard each sample $\boldsymbol{w}^m = \boldsymbol{\mu} + \epsilon_m \boldsymbol{\sigma}$ where $\epsilon_m \sim p(\epsilon)$ is a random draw from some distribution that we take to be an isotropic Gaussian: $p(\epsilon) = \mathcal{N}(0, I)$ with $I$ the $N$-dimensional identity matrix for the $N$ weights of network $W$. These weights $\boldsymbol{w}^m$ are used in the forward pass through the network while parameters $\boldsymbol{\mu}$ and $\boldsymbol{\sigma}$ are trainable. Then, for any function $f(\boldsymbol{w})$ we have $\mathbb{E}_{Q(\boldsymbol{w}|\theta)}[f(\boldsymbol{w})] = \mathbb{E}_{p(\epsilon)}[f(\boldsymbol{w})]$, so that

$$\frac{\partial}{\partial\theta}\mathbb{E}_{Q(\boldsymbol{w}|\theta)}[f(\boldsymbol{w}, \theta)] = \frac{\partial}{\partial\theta}\mathbb{E}_{p(\epsilon)}[f(\boldsymbol{w}, \theta)] = \mathbb{E}_{p(\epsilon)}\left[\frac{\partial f(\boldsymbol{w}, \theta)}{\partial\boldsymbol{w}}\frac{\partial\boldsymbol{w}}{\partial\theta} + \frac{\partial f(\boldsymbol{w}, \theta)}{\partial\theta}\right].$$

The terms are all calculable, allowing us to draw from a distribution for each weight $\boldsymbol{w}^m$ and backpropagate to the underlying distribution parameters $\theta := (\boldsymbol{\mu}, \boldsymbol{\sigma})$. For $\boldsymbol{w}^m = \boldsymbol{\mu} + \epsilon_m \boldsymbol{\sigma}$, the derivatives are $\frac{\partial\boldsymbol{w}^m}{\partial\boldsymbol{\mu}} = I$, and $\frac{\partial\boldsymbol{w}^m}{\partial\boldsymbol{\sigma}} = \epsilon_m I$, making the respective gradients

$$\nabla_{\boldsymbol{\mu}} = \frac{\partial f(\boldsymbol{w}, \boldsymbol{\mu}, \boldsymbol{\sigma})}{\partial\boldsymbol{w}} + \frac{\partial f(\boldsymbol{w}, \boldsymbol{\mu}, \boldsymbol{\sigma})}{\partial\boldsymbol{\mu}} \text{ and } \nabla_{\boldsymbol{\sigma}} = \frac{\partial f(\boldsymbol{w}, \boldsymbol{\mu}, \boldsymbol{\sigma})}{\partial\boldsymbol{w}}\epsilon_m + \frac{\partial f(\boldsymbol{w}, \boldsymbol{\mu}, \boldsymbol{\sigma})}{\partial\boldsymbol{\sigma}}. \tag{11}$$

**PWFN Clustering Algorithm**

**Prior Initialization**

In addition to the prior initialization described in main paper, we added an reweighting determined by the size of the $\sigma$ values in the network. Using the definition of $v = D_{\text{rel}}(\mu_i, \text{pow2}_u(\mu_i))$ we re-weight by the third quartile $\tilde{v}_{0.75}$ and re-write the initialization as:

$$f(\mu_i) = 0.0025 \times \frac{D_{\text{rel}}(\mu_i, \text{pow2}_u(\mu_i)) \times D_{rel}(\mu_i, \text{pow2}_d(\mu_i))}{\tilde{v}_{0.75}}, \tag{12}$$

**while** $|W^{t+1}_{\text{fixed}}| \leq Np_t$ **do**

    $\text{fixed}_{\text{new}} \leftarrow [\,]$

    **while** $\text{fixed}_{\text{new}}$ *is empty* **do**

        Increase the order $\omega \leftarrow \omega + 1$

        Increase $\delta$ $\delta \leftarrow 2\delta$

        $c^\omega \leftarrow \{\sum_{i \in r} i \mid r \in \mathcal{P}(R) \land |r| \leq \omega\}$

        for each $i = 1 \ldots, |W^{t+1}_{\text{free}}|$

            $c^\omega_*(i) \leftarrow \min_{c \in C^\omega} D_{\text{prob}}(w_i, c)$

        for each cluster center $c^\omega_k \in C^\omega$

            $n^\omega_k \leftarrow \sum_i \mathbb{I}[c^\omega_k = c^\omega_*(i)]$

        $k^* \leftarrow \arg\max_k n^\omega_k$

        Sort: $[w'_1, \ldots, w'_N] \leftarrow [w_1, \ldots, w_N]$, $w'_i = w_{\pi(i)}$, $\pi$ permutation

            where $D_{\text{prob}}(w'_i, c^\omega_{k*}) < D_{\text{prob}}(w'_{i+1}, c^\omega_{k*})$

        $i \leftarrow 1$, $\text{mean} \leftarrow D_{\text{prob}}(w'_1, c^\omega_{k*})$

        **while** $\text{mean} \leq \delta$ **do**

            $\text{fixed}_{\text{new}} \leftarrow w'_i$

            $\text{mean} \leftarrow \frac{i}{i+1} \times \text{mean} + \frac{1}{i+1} \times D_{\text{prob}}(w'_{i+1}, c^\omega_{k*})$

            $i \leftarrow i + 1$

        **end**

        $\delta_{t=t} \leftarrow \beta \times \delta_{t=t}$

    **end**

    Assign all the weights means $\mu_i \in \text{fixed}_{\text{new}}$ to cluster center $c^\omega_*(i)$ and set each of the $\sigma_i \in \text{fixed}_{\text{new}}$ to be the variance of the weight means in $\text{fixed}_{\text{new}}$. Finally, move them from $W^{t+1}_{\text{free}}$ to $W^{t+1}_{\text{fixed}}$

**end**

**Algorithm 1:** Clustering $Np_t$ weights at the $t^{th}$ iteration in PWFN.

and clamp the values to be within the range $[2^{-30}, 0.05]$ giving us our initial variance values.

$$\sigma_i = \max(0.1, \min(f(\mu_i), 2^{-30}). \tag{13}$$

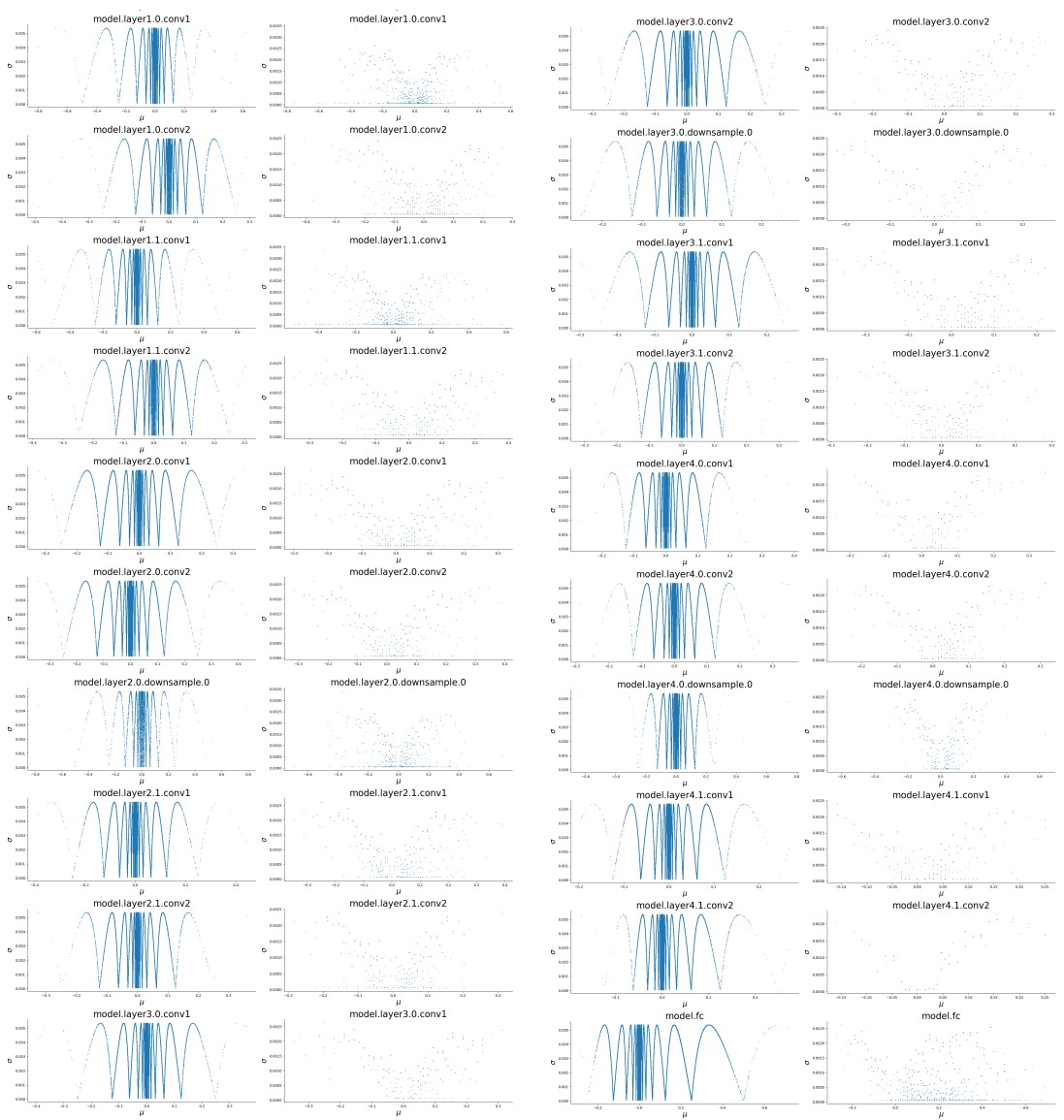

Figure 6: Here we compare the $\mu$ vs $\sigma$ values for all weights in a given layer at initialization (left) and after PWFN convergence and clustering (right).

