# 1  Appendix

**Bayes-by-backprop** The Bayesian posterior neural network distribution $P(\boldsymbol{w}|\boldsymbol{D})$ is approximated by a distribution $Q(\boldsymbol{w}|\theta)$ whose parameters $\theta$ are trained using back-propagation, Bayes-by-backprop (BPP). The approximation is achieved by minimising the Kullback-Leibler (KL) divergence $D_{KL}[Q\|P]$ between $P$ and $Q$ to find the optimal parameters $\theta^*$. These parameters $\theta^*$ instantiate the means $\mu_i$ and variances $\sigma_i^2$ of the PWFN.

$$\theta^* = \arg\min_{\theta} KL[Q(\boldsymbol{w}|\theta)\|P(\boldsymbol{w}|\boldsymbol{D})], \text{ where}$$

$$KL[Q(\boldsymbol{w}|\theta)\|P(\boldsymbol{w}|\boldsymbol{D})] = \mathbb{E}_{Q(\boldsymbol{w}|\theta)} \log\left(\frac{Q(\boldsymbol{w}|\theta)}{P(\boldsymbol{w}|\boldsymbol{D})}\right) = \mathbb{E}_{Q(\boldsymbol{w}|\theta)} \log\left(\frac{Q(\boldsymbol{w}|\theta)P(\boldsymbol{D})}{P(\boldsymbol{D}|\boldsymbol{w})P(\boldsymbol{w})}\right). \tag{1}$$

The $P(\boldsymbol{D})$ term does not contribute to the optimisation and is dropped, leaving

$$\theta^* = \arg\min_{\theta} \mathbb{E}_{Q(\boldsymbol{w}|\theta)}[\log Q(\boldsymbol{w}|\theta) - \log P(\boldsymbol{D}|\boldsymbol{w}) - \log P(\boldsymbol{w})],$$

$$\approx \arg\min_{\theta} \sum_m \underbrace{\log Q(\boldsymbol{w}^m|\theta) - \log P(\boldsymbol{w}^m)}_{\text{prior dependent}} - \underbrace{\log P(\boldsymbol{D}|\boldsymbol{w}^m)}_{\text{data dependent,}} \tag{2}$$

where the expectation value is approximated by samples $\boldsymbol{w}^m \sim Q(\boldsymbol{w}|\theta)$ drawn from $Q(\boldsymbol{w}|theta)$ each of which instantiates a neural network.

$$\mathcal{L}(Q)$$
$$= H(Q(\boldsymbol{w}|\theta)) + \int Q(\boldsymbol{w}|\theta) \log P(\boldsymbol{D}|\boldsymbol{w}) dw$$
$$+ \int Q(\boldsymbol{w}|\theta) \log P(\boldsymbol{w}) dw$$
$$= \int Q(\boldsymbol{w}|\theta) \log P(\boldsymbol{D}|\boldsymbol{w}) dw - \int Q(\boldsymbol{w}|\theta) \log Q(\boldsymbol{w}|\theta) dw$$
$$+ \int Q(\boldsymbol{w}|\theta) \log P(\boldsymbol{w}) dw. \tag{3}$$

Gradient descent over each $\boldsymbol{w}^m$ that instantiates a neural network is made possible by the *reparametrisation trick*. The idea is to regard each sample $\boldsymbol{w}^m = \boldsymbol{\mu} + \epsilon_m \boldsymbol{\sigma}$ where $\epsilon_m \sim p(\epsilon)$ is a random draw from some distribution that we take to be an isotropic Gaussian: $p(\epsilon) = \mathcal{N}(0, I)$ with $I$ the $N$-dimensional identity matrix for the $N$ weights of network $W$. These weights $\boldsymbol{w}^m$ are used in the forward pass through the network while parameters $\boldsymbol{\mu}$ and $\boldsymbol{\sigma}$ are trainable. Then, for any function $f(\boldsymbol{w})$ we have $\mathbb{E}_{Q(\boldsymbol{w}|\theta)}[f(\boldsymbol{w})] = \mathbb{E}_{p(\epsilon)}[f(\boldsymbol{w})]$, so that

$$\frac{\partial}{\partial\theta}\mathbb{E}_{Q(\boldsymbol{w}|\theta)}[f(\boldsymbol{w},\theta)] = \frac{\partial}{\partial\theta}\mathbb{E}_{p(\epsilon)}[f(\boldsymbol{w},\theta)] = \mathbb{E}_{p(\epsilon)}[\frac{\partial f(\boldsymbol{w},\theta)}{\partial\boldsymbol{w}}\frac{\partial\boldsymbol{w}}{\partial\theta} + \frac{\partial f(\boldsymbol{w},\theta)}{\partial\theta}].$$

$$\frac{\partial}{\partial\sigma}\int f(w)Q(w|\theta)dw = \int p(\epsilon)\left[\frac{\partial f(w)}{\partial w}\frac{\partial w}{\partial\sigma} + \frac{\partial f(w)}{\partial\sigma}\right]d\epsilon. \tag{4}$$

The terms are all calculable, allowing us to draw from a distribution for each weight $\boldsymbol{w}^m$ and backpropagate to the underlying distribution parameters $\theta := (\boldsymbol{\mu}, \boldsymbol{\sigma})$. For $\boldsymbol{w}^m = \boldsymbol{\mu} + \epsilon_m \boldsymbol{\sigma}$, the derivatives are $\frac{\partial \boldsymbol{w}^m}{\partial\boldsymbol{\mu}} = I$, and $\frac{\partial \boldsymbol{w}^m}{\partial\boldsymbol{\sigma}} = \epsilon_m I$, making the respective gradients

$$\nabla_{\boldsymbol{\mu}} = \frac{\partial f(\boldsymbol{w},\boldsymbol{\mu},\boldsymbol{\sigma})}{\partial\boldsymbol{w}} + \frac{\partial f(\boldsymbol{w},\boldsymbol{\mu},\boldsymbol{\sigma})}{\partial\boldsymbol{\mu}} \text{ and } \nabla_{\boldsymbol{\sigma}} = \frac{\partial f(\boldsymbol{w},\boldsymbol{\mu},\boldsymbol{\sigma})}{\partial\boldsymbol{w}}\epsilon_m + \frac{\partial f(\boldsymbol{w},\boldsymbol{\mu},\boldsymbol{\sigma})}{\partial\boldsymbol{\sigma}}. \tag{5}$$

where $w^i$ corresponds to the $i$th sample drawn from the variational posterior $Q(w^i|\theta)$. We can define $f(\boldsymbol{w},\boldsymbol{\theta}) = \log Q(\boldsymbol{w}|\theta) - \log P(\boldsymbol{w}) - \log P(\mathcal{D}|\boldsymbol{w})$ and update using gradient descent.

Using $\theta = (\boldsymbol{\mu}, \boldsymbol{\sigma})$ we have that:

$$\Delta_{\mu} = \frac{\partial f(\boldsymbol{w},\boldsymbol{\mu},\boldsymbol{\sigma})}{\partial\boldsymbol{w}} + \frac{\partial f(\boldsymbol{w},\boldsymbol{\mu},\boldsymbol{\sigma})}{\partial\boldsymbol{\mu}}, \tag{6}$$

24 and:

$$\Delta_\sigma = \frac{\partial f(\boldsymbol{w}, \boldsymbol{\mu}, \boldsymbol{\sigma})}{\partial \boldsymbol{w}} \epsilon + \frac{\partial f(\boldsymbol{w}, \boldsymbol{\mu}, \boldsymbol{\sigma})}{\partial \boldsymbol{\sigma}}. \tag{7}$$

25

## PWFN Clustering Algorithm

27 In Algorithm 1 we give the full clustering algorithm used for each of the $T$ fixing iterations.

> **while** $|W_{\text{fixed}}^{t+1}| \leq N p_t$ **do**
> > $\omega \leftarrow 0$
> > $\text{fixed}_{\text{new}} \leftarrow [\,]$
> > **while** $\text{fixed}_{\text{new}}$ *is empty* **do**
> > > Increase the order: $\omega \leftarrow \omega + 1$
> > > $c^\omega \leftarrow \{\sum_{i \in r} i \mid r \in \mathcal{P}(R) \wedge |r| \leq \omega\}$
> > > for each $i = 1 \ldots, |W_{\text{free}}^{t+1}|$
> > > $\quad c_*^\omega(i) \leftarrow \min_{c \in C^\omega} D_{\text{prob}}(w_i, c)$
> > > for each cluster centre $c_k^\omega \in C^\omega$
> > > $\quad n_k^\omega \leftarrow \sum_i \mathbb{I}[c_k^\omega = c_*^\omega(i)]$
> > > $k^* \leftarrow \arg\max_k n_k^\omega$
> > > Sort: $[w_1', \ldots, w_N'] \leftarrow [w_1, \ldots, w_N]$, $w_i' = w_{\pi(i)}$, $\pi$ permutation
> > > $\quad$ where $D_{\text{prob}}(w_i', c_{k*}^\omega) < D_{\text{prob}}(w_{i+1}', c_{k*}^\omega)$
> > > $i \leftarrow 1$, $\text{mean} \leftarrow D_{\text{prob}}(w_1', c_{k*}^\omega)$
> > > **while** $\text{mean} \leq \delta$ **do**
> > > > $\text{fixed}_{\text{new}} \leftarrow w_i'$
> > > > $\text{mean} \leftarrow \frac{i}{i+1} \times \text{mean} + \frac{1}{i+1} \times D_{\text{prob}}(w_{i+1}', c_{k*}^\omega)$
> > > > $i \leftarrow i + 1$
> > > 
> > > **end**
> > > $\delta \leftarrow 2 \times \delta$
> > 
> > **end**
> > Assign all the weights means $\mu_i \in \text{fixed}_{\text{new}}$ to cluster centre $c_*^\omega(i)$ and set each of the
> > $\quad \sigma_i \in \text{fixed}_{\text{new}}$ to be the variance of the weight means in $\text{fixed}_{\text{new}}$. Finally, move them
> > $\quad$ from $W_{\text{free}}^{t+1}$ to $W_{\text{fixed}}^{t+1}$
> 
> **end**

**Algorithm 1:** Clustering $N p_t$ weights at the $t^{th}$ iteration in PWFN.

### 1.1 Prior Initialisation

29 In addition to the prior intialisation described in main paper, we added an reweighting determined by
30 the size of the $\sigma$ values in the network. Using the definition of $v = D_{\text{rel}}(\mu_i, \text{pow2}_u(\mu_i))$ we re-weight
31 by the third quartile $\tilde{v}_{0.75}$ and re-write the initialisation as:

$$f(\mu_i) = 0.0025 \times \frac{D_{\text{rel}}(\mu_i, \text{pow2}_u(\mu_i)) \times D_{rel}(\mu_i, \text{pow2}_d(\mu_i))}{\tilde{v}_{0.75}}, \tag{8}$$

32 and clamp the values to be within the range $[2^{-30}, 0.05]$ giving us our initial variance values.

$$\sigma_i = \max(0.1, \min(f(\mu_i), 2^{-30})). \tag{9}$$

33 In Figure 1 we show how the layers' $\sigma$ and $\mu$ values are initialised using the prior (left) and where
34 they converge to (right) given a ResNet-18 model trained on the ImageNet dataset.

## 2 Hyper Parameter Search

36 We conduct an extensive hyperparameter search looking at combinations of $\alpha$, the number of training
37 epochs between rounds, and the $\gamma$ threshold on the Cifar10 dataset and Resnet18 model.

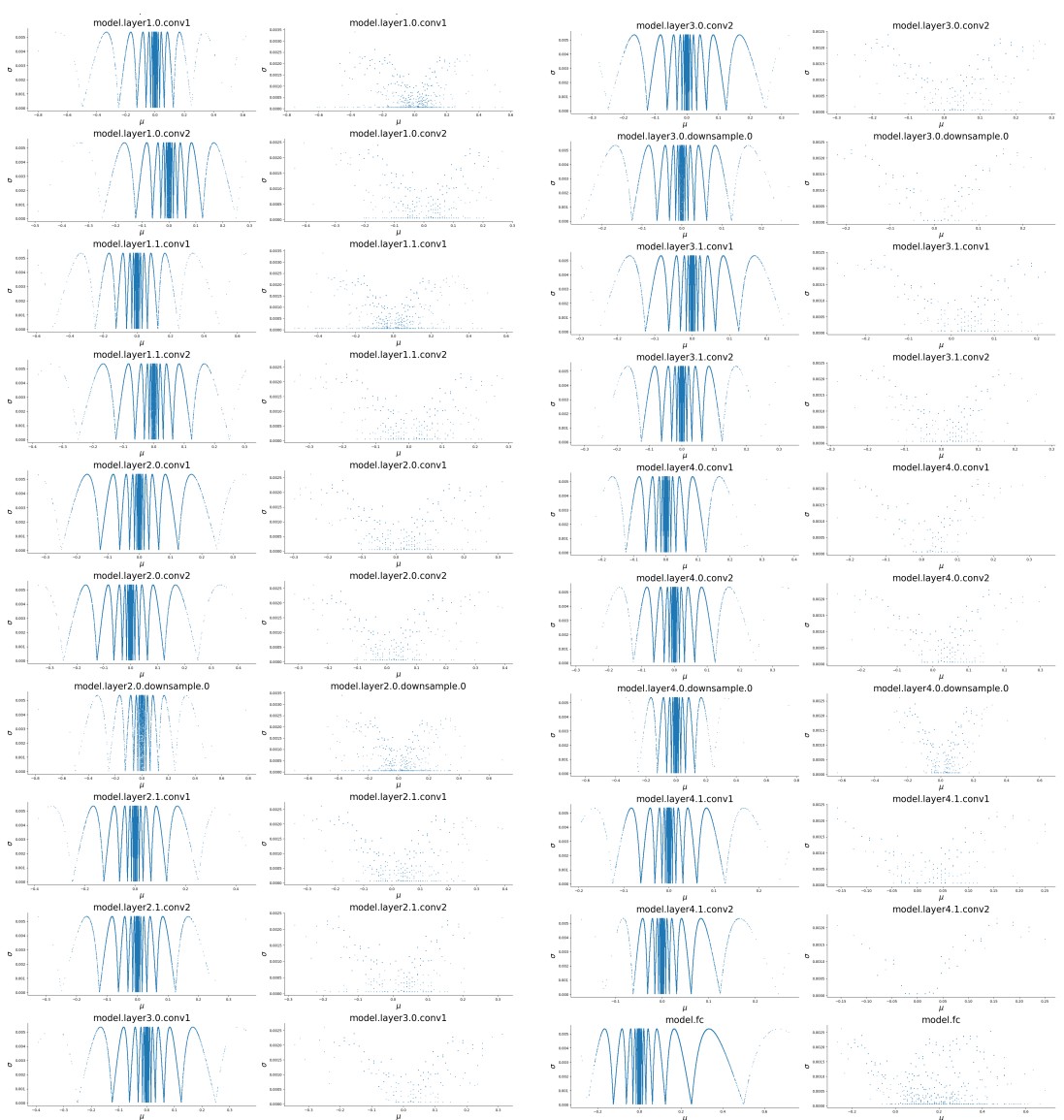

Figure 1: Here we compare the $\mu$ vs $\sigma$ values for all weights in a given layer at initialisation (left) and after PWFN convergence and clustering (right).

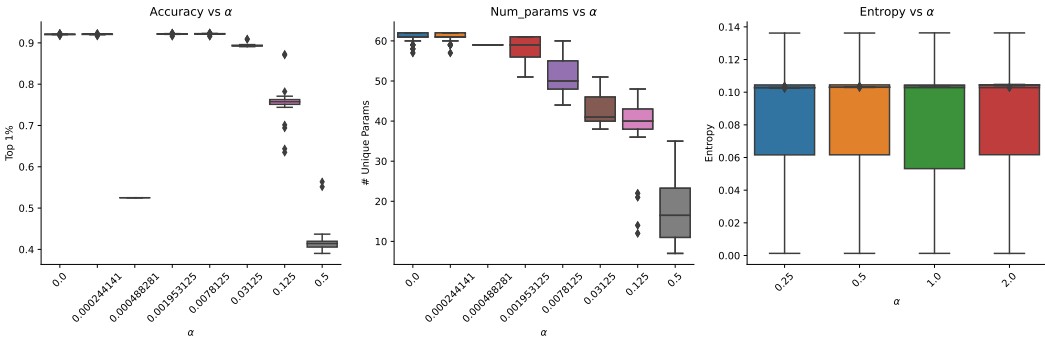

Figure 2: Here we show how the $\alpha$ regulariser impacts accuracy and compressibility.

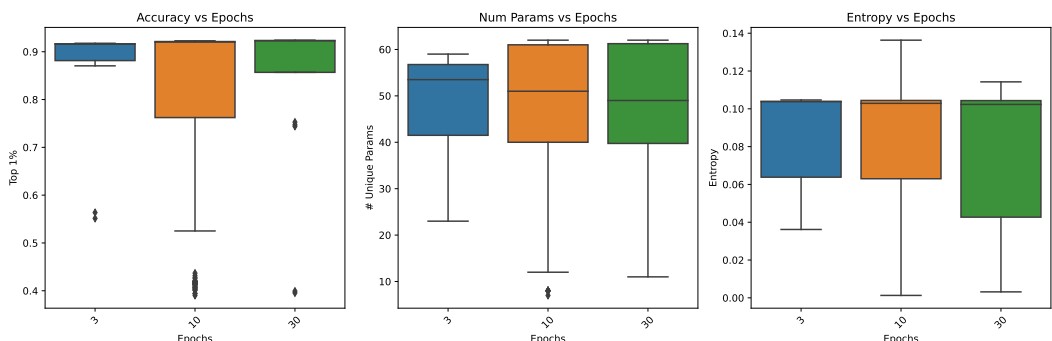

Figure 3: Here we see that only a few epochs are needed to maintain accuracy between clustering stages

In Figure 2 we show the impact of increasing the regularisation strength. In Figure 3 we see that only
3 epochs is necessary to maintain accuracy and strong compressibility. Finally, in Figure 4 we see
how the accuracy, number of unique parameters and weight-space entropy changes across all the
hyperparameter combinations explored.

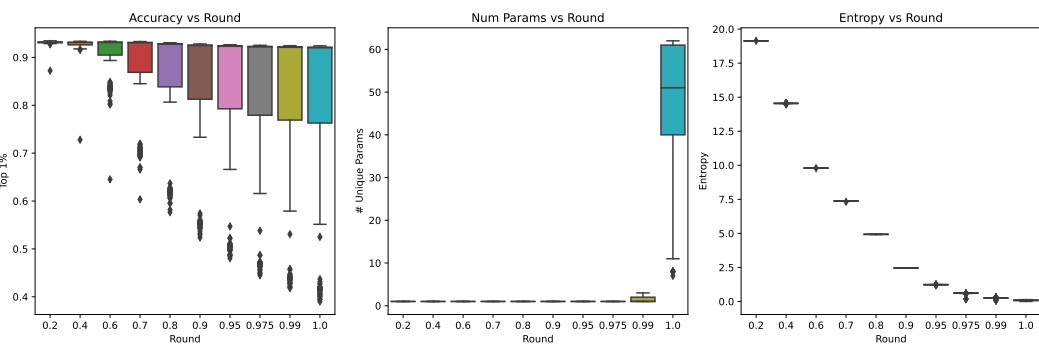

Figure 4: Here we see how the accuracy, number of unique parameters and weight space entropy
evolves over each weight fixing round