# OpenReview forum: "Probabilistic Weight Fixing: Large-scale training of neural network weight uncertainties for quantisation."
_NeurIPS.cc/2023/Conference — NeurIPS 2023 poster_

### Official Review · Reviewer_jtUH · 2023-07-02

**Soundness:** 3 good
**Presentation:** 2 fair
**Contribution:** 2 fair
**Rating:** 5
**Confidence:** 4

**Summary:**

This paper discusses a method of quantizing weights based on Bayesian neural networks (BNN) by clustering them as closely as possible to a set of powers-of-two values. Unlike the weight fix network (WFN) that quantizes by clustering the fixed weights of the network around the nearest centroid, the proposed Probabilistic weight fix network (PWFN) makes use of a probabilistic BNN model. Weights are assumed to follow a gaussian distribution. The $\mu$'s and $\sigma$'s of the weights are learned using Bayes-by-backprop (BBP). Quantization is performed iteratively by clustering around a specific centroid using the learned $\mu$ and  $\sigma$ from the model’s codebook.

**Strengths:**

By expressing the existing network as a weight-shared BNN, several advantages can be obtained:
- It allows for uncertainty quantification.
- Using the Mahalanobis distance to determine the centroid seems more reasonable than using the Euclidean distance or Relative movement distance.

In many existing papers, codebooks were maintained at the layer or channel level, requiring significant storage capacity. However, this paper achieves higher quantization efficiency by utilizing a codebook at the model level.

**Weaknesses:**

- The amount of computation is significantly increased compared to the conventional WFN. It is also harder to train than a regular network, which also increases the time for hyperparameter exploration.

- Because of its complexity (BNN), it is unclear whether it can be applied to large models.

- The choice of prior and the learning of mu and sigma can greatly influence the results. This can lead to inconsistent performance with each training and cases where performance drops significantly. Using ensembles seems like it would require too much inference time.

- Sigma is set as the distance from the actual parameter to the power of two, but this could be different from the actual sensitivity of the parameter and could be a wrong choice of prior.

- Reference format should be easily distingusiable from that of equation numbering. Since both uses parenthesis (), it looks quite odd at first sight.

- According to the NeurIPS 2023 guide, all table numbers and titles should always appear before the table. https://media.neurips.cc/Conferences/NeurIPS2023/Styles/neurips_2023.pdf

- Some symbols in the equations are insufficiently explained regarding their dimension, type, or meaning. Line 192 Θ, Line 193 S, Line 201 δ. Also, δ is duplicated with the threshold δ on Line 237.

**Questions:**

- Q1) How was 0.05 determined in equation 4? Is it a value that can be generally applied to all networks, or is it a hyperparameter that should be applied differently for each model?

- Q2) Can you provide the standard deviation of accuracy in ensemble models? And can you provide the maximum and minimum accuracy of training?

- Q3) How is the Top-1 accuracy in Table 1 determined, since the results may vary for each training?

- Q4) It seems like using BNN would greatly increase training time; how much? Can you tell us how much the training time increases for ResNet-18 and DeiT compared to WFN?

- Q5) How much time/MACs are required for training and inference?

**Limitations:**

Limitations are not discussed.

---

> ### Author Rebuttal · Authors · 2023-08-04
>
> We appreciate the comprehensive feedback provided on our paper. Herein, we address your concerns systematically:
>
> **Q1) The Determination of 0.05 in Equation 4**
>
> The value 0.05 was empirically determined through experiments on Cifar-10 using ResNet18. While this parameter demonstrated consistent performance across our tested networks, further exploration may optimize it for different architectures.
>
> **Q2)** Can you provide the standard deviation of accuracy in ensemble models? And can you provide the maximum and minimum accuracy of training?
>
> Thank you for highlighting the importance of this metric. Here are the detailed results:
>
> | Model       | Max   | Min   | Std  |
> |-------------|-------|-------|------|
> | ResNet18    | 70.23 | 69.99 | 0.08 |
> | ResNet34    | 74.99 | 74.34 | 0.16 |
> | ResNet50    | 79.08 | 77.33 | 0.42 |
> | DeiT-Small  | 78.86 | 78.00 | 0.23 |
> | DeiT-Tiny   | 71.79 | 71.38 | 0.17 |
>
>
> **Q3)** How is the Top-1 accuracy in Table 1 determined, since the results may vary for each training?
>
> For the Top-1 accuracy, we fixed the quantized weights to their means (mu values). PWFN allows for both a compressed network for efficient inference and, optionally, an ensemble for uncertainty estimation.
>
>
> **Q4)** It seems like using a BNN would greatly increase training time; how much? Can you tell us how much the training time increases for ResNet-18 and DeiT compared to WFN?
>
> Indeed, BNN adds an overhead, but it's manageable with current GPU resources. PWFN regularization also lessens the sigma-collapse challenge, making it feasible even for larger models. We've added a general rebuttal comment to further elaborate.
>
>
> **Q5)** How much time/MACs are required for training and inference?
>
> For inference in the quantized network, the time matches other quantized networks of similar compression rates. Uncertainty estimations introduce additional costs but only if deemed beneficial by the practitioner. These costs are comparable to running inference multiple times on the same network (aside from the random sampling step), and will likely have much less overhead when compared to ensembling separately trained models.
>
> **On the Complexity of BNNs**
>
> As we show in the general comment, the overhead introduced training a BNN is manageable on consumer GPUs when using the PWFN regularisation term to avoid sigma-collapse. This means that PWFN can indeed be applied to large models. As far as we are aware, we are the first to successfully train a BNN on imagenet-scale problems, demonstrating strong performance on large and small models from the ResNet family as well as vision transformer models, and we see no reason why this can not be scaled further, we're excited to see how far training BNNs in this way can be pushed forward in future works.
>
> **On Prior Choice and Consistency of Performance**
>
> While different priors might influence results, our choice is backed by previous works and targets optimal compression for hardware accelerators. The notion of inconsistency has been addressed in our response to Q2.
>
> **Clarifications on Symbols**
>
>     • Θ – this is the Heaviside step function. defined on Line 192.
>
>     • S – a constant as defined in Lines 194-196 and equation 3.
>
>     • δ – We've revised the text to clearly differentiate the context of its multiple usages.
>
> **Choice of Prior**
>
> While we acknowledge that other priors might optimize performance further, our choice is informed by previous works and the hardware efficiencies of mapping weights to powers-of-two.
>
> **Table Formatting**
>
> We thank you for pointing out the format discrepancy. It's now corrected as per the NeurIPS 2023 guide.
>
> **Conclusion**
>
> We value the feedback and believe the clarifications offered reinforce the strength and novelty of our methodology. We remain hopeful that our approach's advantages, as detailed in the paper and this rebuttal, and the utility of the work to the wider community can be seen.

---

### Official Review · Reviewer_1QKt · 2023-07-06

**Soundness:** 3 good
**Presentation:** 3 good
**Contribution:** 2 fair
**Rating:** 5
**Confidence:** 2

**Summary:**

PWFN (Probabilistic Weight-Shared Fusion Network) is a technique that combines weight-sharing quantization with Bayesian neural networks to achieve highly compressed and quantized neural networks. It models each weight as a draw from a distribution, allowing for the quantification of uncertainty in weight values. PWFN consists of iterative stages of training and clustering to optimize the weight distributions and determine optimal cluster configurations.

PWFN offers a probabilistic approach to weight-sharing quantization, providing enhanced compressibility, flexibility, and noise resilience in neural networks. It allows for uncertainty quantification and demonstrates improved performance compared to existing methods.

**Strengths:**

(1)  By representing weights as probability distributions, PWFN provides flexibility in weight values and enhances noise resilience. The distributions capture uncertainty in weight values, enabling robust performance even in the presence of noise.

(2) PWFN incorporates a regularization term and iterative clustering procedure, leading to improved performance compared to other methods

(3) By sampling weights from their distributions and observing changes in predictions, the model's confidence and uncertainty can be assessed.

**Weaknesses:**

(1) PWFN needs additional complexity in the training process including regularization terms and clustering iterations.

(2) PWFN may require modifications to existing neural network frameworks to incorporate the probabilistic weight-sharing framework. It may also require additional expertise in Bayesian neural networks and variational inference techniques. For the reproducibility, the code needs to be open.

(3) Modeling weights as distributions and performing sampling during training can introduce additional computational overhead compared to traditional point estimate-based methods. This may result in increased training time and inference latency.

**Questions:**

See Weaknesses

---

> ### Author Rebuttal · Authors · 2023-08-04
>
> Thank you for your insightful comments and constructive feedback on our submission. We appreciate the time and effort you've taken to review our paper.
>
> **Complexity in Training Process**: We understand your concern about the additional complexity introduced by PWFN during the training process and have added a general comment with regards to this which we hope alleviates your concerns.
>
> **Framework Modifications and Expertise**: We're pleased to share that we'll be releasing the code, and the process of mapping a traditionally trained network to a BNN will be automated - a user need only pass to the code a pre-trained model. This will significantly reduce the barriers to adoption, allowing those unfamiliar with Bayesian neural networks or variational inference techniques to still benefit from PWFN.
>
> **Computational Overhead**: You rightly pointed out the potential computational overhead when modeling weights as distributions. However, it's crucial to note that this overhead is mainly seen during training and isn't any larger than existing re-training quantization approaches.
> For standard inference tasks, our method uses quantized point estimates for weights, ensuring no added latency.
> When leveraging the BNNs for uncertainty estimation there are additional costs, but these are comparable to running inference multiple times on the same network (aside from the random sampling step), and will likely have much less overhead when compared to ensembling separately trained models.
>
> Once again, thank you for your review, we hope our revisions address your concerns adequately.

---

### Official Review · Reviewer_tpmz · 2023-07-07

**Soundness:** 2 fair
**Presentation:** 2 fair
**Contribution:** 3 good
**Rating:** 6
**Confidence:** 3

**Summary:**

This paper discusses the Weight-sharing quantization technique to reduce DRAM read costs. To address this issue, an iterative weight fixing scheme was employed, which involved alternating between network training and weight clustering. This paper proposes a BNN-based framework that utilizes a new initialization setting and regularization term. With this method, millions of weights can be represented by hundreds of unique values.

**Strengths:**

Probabilistic approach for weight-sharing quantization is novel and interesting.
The method improves the accuracy while minimizing the entropy and reducing the number of unique values used to represent the weights.
The prior initialization method using relative distance from powers-of-two consistently improves the accuracy, especially on resnet architectures.

**Weaknesses:**

There is no explanation in the manuscript about how the weight position described in the abstract was handled.
Lack of experimental support for regularization scheme (x)
Lack of clarification – No explanation about Figure 2 and 3 in the manuscript

**Questions:**

(49) There is no explicit clarification for the abbreviation of PWFN.
There is no explanation for Figure 2 and 3 in the manuscript.
(220) what is the meaning of j index?

**Limitations:**

The paper does not address the limitation of the method and any the negative societal impact.

---

> ### Author Rebuttal · Authors · 2023-08-04
>
> We greatly appreciate the feedback and constructive criticisms provided by the reviewer. Your insights are invaluable to improving our manuscript. Here, we address the concerns raised:
>
> **Concern 1: Weight Position Handling**
>
> The weight position is indeed utilised in our methodology when determining the cluster assignments. As you rightly pointed out, each weight is associated with an individual sigma value that aids in determining its cluster assignment. This unique approach ensures that weights, even with the same value but in different positions, can be assigned to distinct clusters. We realized the need for more clarity in our manuscript regarding this aspect and have amended the methodology section to elucidate this further.
>
> **Concern 2: Lack of Experimental Support and Clarification**
>
> Thank you for pointing out the ambiguities associated with Figures 2 and 3. We've now expanded the captions for both figures to provide clarity. Specifically, Figure 2 showcases how increased alpha value (the scaling of the regularization term) plays a pivotal role in preventing sigma values from collapsing to zero. Avoiding this collapse is important, both in terms of using the network as a Bayesian Neural Network (if sigma is zero, then we have a point-estimate) and in our downstream use of using sigma to determine the quantization mapping. We hope this clarifies that experiments have been conducted and presented to explore the value of the regularization term in the methodology. Figure 3 has also been revised to enhance readability and comprehension.
>
> **Concern 3: Clarification on Abbreviations**
>
> Our sincere apologies for the oversight. We have now clearly defined the abbreviation as "Probabilistic Weight Fixing Networks (PWFN)" in the relevant sections of the manuscript.
>
> **Conclusion**
>
> We believe our work on the bayesian weight-sharing quantization technique is a valuable contribution to the field, and we're confident that the changes we've made, following your feedback, will further clarify our methodology and its significance. Once again, thank you for your constructive feedback, and we hope our revisions address your concerns adequately.

---

> > ### Comment · Reviewer_tpmz · 2023-08-18
> >
> > As the authors reasonably address our concerns and issues, there is no further requests except for adding the detailed explanation of figure 2 and 3 in the main body of the manuscript by citing them instead of adding it in the captions.

---

> > > ### Author Response · Authors · 2023-08-18
> > >
> > > That's great - thank you for clarification on this, we'll add the references to the figures in the main body as you suggest.

---

### Official Review · Reviewer_NcTB · 2023-07-08

**Soundness:** 3 good
**Presentation:** 3 good
**Contribution:** 3 good
**Rating:** 7
**Confidence:** 3

**Summary:**

This paper presents a novel quantization scheme based on iterative training and clustering of the weights into a very limited and finite set of choices. The paper follows a Bayesian approach to assign weights to clusters. Experiments are demonstrated on ImageNet for ResNet and Transformer architectures.

**Strengths:**

The paper addresses an important topic of improving quantization. The paper is very clearly written, easy to understand and the idea is simple and elegant. Experiments seem sufficient to demonstrate the power of the method.


**Weaknesses:**

 I do not see any immediate weakness of the paper. One thing which could make the paper stronger is to explore the application on more complicated downstream tasks of for example DeiT or DeiT-small. This is just to understand the scalability of the method. Further, authors could also provide details on the added training costs to also challenges faced in the training involved for the approach. Additionally, detailed ablation is missing on how sensitive the results are w.r.t. the chosen prior, number of training+clustering combination steps, etc.

**Questions:**

See weaknesses reported above.

**Limitations:**

Limitation in the context of hardware has been stated by the authors. However, it would be nice to understand additional limitations w.r.t. training time compared to other quantization methods, and other features that play role in the method.

---

> ### Author Rebuttal · Authors · 2023-08-04
>
> Thank you for your thoughtful review and valuable insights on our paper.
>
> **Training Costs**: We appreciate the concern raised about the training costs. Given its recurrent mention in the reviews, we've expanded on this aspect, providing a comparative analysis between PWFN and other prevalent quantization approaches as general comment. We hope this will shed light on the efficiency of our method in terms of training overheads.
>
> **Number of Training+Clustering Steps**: Your observation on the potential optimization of the number of training+clustering steps is apt. While we initially set the number of weight-fixing steps based on the success of WFN, we recognize the scope for fine-tuning. Post your review, we experimented by reducing the steps (down to 7) for CIFAR-10, which yielded encouraging results. However, it remains to be seen if this can be generalized to larger datasets. We acknowledge this as a potential area for future investigation.
>
> **Expansion to Other Domains**: We concur with your suggestion on diversifying the application domains. Our focus on image classification was largely inspired by preceding works, but we did venture into ViT-type architectures. Notably, with Bayesian Networks, PWFN stands out as the first approach to train image-net scale models. As for quantization, PWFN is one of only a few methods to successfully quantize DeiT-family models.
>
> **Future Research on BNN Quantization**: Your idea of expanding the scope of BNN Quantization to a broader range of tasks is indeed promising. We hope our work paves the way for subsequent research endeavors in this direction.
>
> Thank you again for your constructive feedback.

---

> > ### Comment · Reviewer_NcTB · 2023-08-19
> > **Thank you for the response**
> >
> > Thanks for responding with the explanation. I am convinced and I retain the rating.

---

### Official Review · Reviewer_TzYT · 2023-07-09

**Soundness:** 2 fair
**Presentation:** 2 fair
**Contribution:** 2 fair
**Rating:** 5
**Confidence:** 4

**Summary:**

This paper introduces a novel approach to weight-sharing quantization, a technique aimed at reducing energy costs associated with inference in deep neural networks. The authors propose a method that takes into account the context of weights in the network, arguing that this strategy can better preserve the network's representational capacity while reducing its complexity. They employ a probabilistic framework to capture the flexibility in weight values, which guides clustering decisions to reduce the network's entropy and lower the unique parameter count without compromising performance. The authors also suggest a novel initialization setting and a regularization term to prevent the collapse of the weights' distribution variance to zero. The paper demonstrates superior compressibility and accuracy in ResNet family models trained on the ImageNet dataset and transformer-based architectures.

**Strengths:**

To the best of my knowledge, this paper is the first to utilize Bayesian Neural Networks for model quantization.

The authors conduct experiments on powers-of-two and additive powers-of-two quantization, which are more hardware-friendly than other types of quantization.

**Weaknesses:**

The paper's writing style is somewhat difficult to follow, particularly in the method presentation and the explanation of Table 1 (In-ch?).

I recommend the authors use the term 'Quantization' instead of 'Quantisation' for consistency with most literature in the field.

The authors should address the training difficulty associated with Bayesian Neural Networks and discuss whether this approach is simpler for model quantization compared to other methods.

**Questions:**

See Weaknesses

**Limitations:**

See Weaknesses

---

> ### Author Rebuttal · Authors · 2023-08-04
>
> First and foremost, we genuinely appreciate your constructive feedback on our paper. It is our aim to present our research in the clearest possible manner, and your insights are invaluable in this pursuit.
>
> **Bayesian Neural Networks (BNNs) for Model Quantization**: You're right in pointing out the uniqueness of utilizing BNNs for model quantization. We feel it is essential to highlight this, and we're pleased that you recognized its novelty.
>
> **Explanation of Table 1**: We agree that a more explicit explanation of Table 1 is needed and have amended this. in-ch refers to whether the approach uses a different codebook for each channel. So if we had a layer with weights with the dimensions: (in-ch, out-ch, filter-width, filter-height) then for each in-ch there would be a different set of unique weights used in the quantisation procedure. This both increases the complexity of inference (usually in the form of different scale/shift constants) and increases the memory requirements.
>
>    **Consistency in Terminology**: We had used the British spellings throughout the paper, but will swap this over to the American spelling to be consistent with the literature.
>
> **Training Costs of BNNs**: Given that multiple reviewers have questions around the training costs associated with Bayesian Neural Networks, we've incorporated a response addressing this aspect in the general rebuttal comments.
>
> Once again, thank you for your review, we hope our revisions adequately address your concerns.

---

### Author Rebuttal · Authors · 2023-08-04

### General Comments

Thank you to all the reviewers for their effort in reviewing our paper, kind comments, and suggestions for improvement. We have taken the comments and suggestions into consideration.

As a general confirmation, it is true that our work is the first to train Bayesian Neural Networks (BNNs) for the purpose of Quantization. To the best of our knowledge, we can make the even stronger claim that we are the first to train a BNN on ImageNet and achieve results that match (or exceed) standard training for the ResNet family of models, and exceed the SOTA accuracies for quantized transformer-based vision models.

Before addressing individual comments, we wanted to address a shared concern around the training costs incurred with training a BNN using the PWFN method.

### Training Costs

It is indeed true that PWFN incurs additional training compared to the original baseline, but when we compare it to other post-training quantization methods:

| Method | Num of additional epochs |
|--------|--------------------------|
| ApoT   | 120                      |
| PWFN   | 27                       |
| WFN    | 27                       |
| LSQ    | 90                       |
| QviT   | 300                      |


We can see that PWFN requires substantially fewer additional training epochs than all methods but WFN.

WFN uses a regularizer that calculates the relative distance between all free weights and the existing cluster centers, and then penalizes weights depending on the distance to their closest center (in a soft way). This incurs computational costs in the backpropagation calculation for every iteration. In PWFN, we have a much simpler regularization term that penalizes sigma to increase - with costs that match that of l0 regularization. We do have memory overhead in terms of the number of parameters at training (sigma and mu) and the random number generation to sample, but this is only at training time. The simplicity of the regularization term also means that we experience a speed-up over the previous BNN training procedure outlined in the original bayes-by-backprop paper, making much more complex model-dataset pairings tractable.

This is not to say there are no costs; we find that a single training epoch with ResNet-18 on the ImageNet dataset takes ~1 hour 30mins on 4 consumer GPUs (GTX1080's) for PWFN, compared with 40 minutes for standard training and ~1 hour 20mins for WFN (but for WFN this increases as the number of clusters increases through training).

### Inference

At inference, the weights are fixed to mu values and treated in the usual way; there are no additional computation costs, and quantization speed-up on supported accelerators can be achieved. The core idea of PWFN is to reduce the number of unique weights and weight-space entropy so that the resultant networks can be used on accelerators (such as EIE) which use Huffman coding as a compression scheme.

### Key Points

Re-highlighting the core contributions of this work:

* This is the first work to train BNNs on imagenet-scale problems
* PWFN reduces the param count and entropies of transformer-based architectures with SOTA quantization performance
* We achieve this with 27 epochs of additional training
* Mapping to a BNN allows for both highly compressed networks and uncertainty estimation probing if required through sampling
* The point-estimates can be used in accelerator designs in conjunction with Huffman encoding to reduce computationally expensive DRAM reads
* We use a single codebook for the entire network whereas most previous works needed multiple codebooks for each layer/in-channel/attention head to maintain performance

Thank you again for taking the time to review our work and we hope to have covered all your points in the individual responses.

---

### Decision · Program_Chairs · 2023-09-21

**Decision:**

Accept (poster)

**Comment:**

Author rebuttal addressed most of the reviewer concerns and all reviewers are leaning accept after final discussion.